# Effect of Internal Curing by Super Absorbent Polymer on the Autogenous Shrinkage of Alkali-Activated Slag Mortars

**DOI:** 10.3390/ma13194318

**Published:** 2020-09-28

**Authors:** Pengju Wang, Haiming Chen, Peiyuan Chen, Jin Pan, Yangchen Xu, Hao Wang, Wenfeng Shen, Ke Cao

**Affiliations:** 1School of Civil Engineering and Architecture, Anhui University of Science and Technology, Huainan 232001, China; wpj15955490880@163.com (P.W.); 17855475408@163.com (J.P.); 2018200261@aust.edu.cn (Y.X.); 2018200211@aust.edu.cn (H.W.); 2018200207@aust.edu.cn (W.S.); 2018200185@aust.edu.cn (K.C.); 2Engineering Research Center of Underground Mine Construction, Ministry of Education, Anhui University of Science and Technology, Huainan 232001, China

**Keywords:** alkali activated slag, super absorbent polymer, autogenous shrinkage, compressive strength, microstructure

## Abstract

Alkali activated slag (AAS) mortar is becoming an increasingly popular green building material because of its excellent engineering properties and low CO_2_ emissions, promising to replace ordinary Portland cement (OPC) mortar. However, AAS’s high shrinkage and short setting time are the important reasons to limit its wide application in engineering. This paper was conducted to investigate the effect of internal curing(IC) by super absorbent polymer (SAP) on the autogenous shrinkage of AAS mortars. For this, an experimental study was carried out to evaluate the effect of SAP dosage on the setting time, autogenous shrinkage, compressive strength, microstructure, and pore structure. The SAP were incorporated at different dosage of 0, 0.05, 0.1, 0.2, 0.3, 0.4, and 0.5 percent by weight of slag. The workability, physical (porosity), mechanical, and shrinkage properties of the mortars were evaluated, and a complementary study on microstructure was made. The results indicated that the setting time increased with an increase of SAP dosage due to the additional activator released by SAP. Autogenous shrinkage decreased with an increase of SAP dosage, and was mitigated completely when the dosage of SAP ≥ 0.2% wt of slag. Although IC by means of SAP reduced the compressive strength, this reduction (23% at 56 days for 0.2% SAP) was acceptable given the important role that it played on mitigating autogenous shrinkage. In the research, the 0.2% SAP dosage was the optimal content. The results can provide data and basis for practical application of AAS mortar.

## 1. Introduction

As a basic building material, concrete has greatly promoted the development of the construction industry, and it will continue to be in demand far into the future [1,2]. However, the Portland cement industry is known to be a consumer of large amounts of energy (3100–3600 kJ/kg) [3], and the production of one ton of ordinary Portland cement (OPC) will release about one ton of CO_2_, which is not friendly to the environment and biodiversity [4]. Recent years, great efforts have been paid on seeking alternative binders to OPC towards to developing in a sustainable way. Those alternative green binders include geopolymer, Alkali activated slag(AAS) sulphate aluminium cement, and magnesium phosphate cement, which provide the concrete industry many chances to develop in more sustainable ways [5,6]. Among them, AAS is such a promising alternative binder with features like quick set, high early-age strength, good durability, low hydration heat, high resistance to chemical attack, etc. [7,8,9]. Duxson et al. [10] proposed that the discharge of CO_2_ can be reduced as high as 80% when replacing OPC totally by AAS. Jiang et al. [11] pointed out that AAS concrete saved 43% more energy and reduced CO_2_ emissions by 73% compared to OPC concrete.

Although researchers have been motivated to study the chemical and physical properties of AAS in the last 30 years, the successful application of such a green binder is seldom reported. One of the major reasons is that AAS normally suffers to excessive autogenous shrinkage during its hardening process. For example, Cartwright et al. [12] found that the autogenous shrinkage of AAS mortar was 6 times larger than that of OPC mortar. Chen et al. [13] even confirmed that the autogenous shrinkage of AAS mortars was as high as 30–60 times larger than that of OPC mortars depending on activators, and the 28 day autogenous shrinkages of AAS mortars were several thousand micro strain. Considering the fact that the ultimate tension strain of a normal concrete is 75–115 µε, AAS mortar or concrete is vulnerable to crack during its service life. Therefore, without effective mitigation of such autogenous shrinkage of AAS, it can’t be applied into practical projects.

On the mechanism of autogenous shrinkage, it is aroused by the capillary tension of unsaturated pores due to self-desiccation [14]. Supplying additional water internally to promote the relative humidity of pores has been proved as an effective method, which is also well known as IC. IC was originally defined by the American Concrete Institute (ACI) as supplying water throughout a freshly placed cementitious mixture using reservoirs, via pre-wetted lightweight aggregates [15], pre-wetted crushed returned concrete fines, super absorbent polymer(SAP) and other carriers that readily release water, as needed for hydration or to replace moisture lost through evaporation or self-desiccation. SAP is widely used for IC of cement-based materials: pastes [16,17], mortars [18,19], concretes [20,21,22,23,24,25,26], and alkali-activated materials: pastes [27,28], mortars [29,30,31,32,33] due to its super water absorption capacity. Oh and Choi [30] investigated the effect of SAP on the hydration process and autogenous shrinkage. They found that by adding 0.3% SAP the autogenous shrinkage of AAS mortars can be maximally reduced by 72.7%. Tu et al. [27] studied the relationship between SAP dosage and the autogenous shrinkage of AAS mortars as well as the IC efficiency. Vafaei et al. [28] confirmed the efficiency of IC by SAP on the performance of AAS mortars.

However, the effect of SAP as IC agent on shrinkage of AAS mortar have not been intensively investigated. The relationship between SAP dosage and autogenous shrinkage of AAS mortars has not been established. More studies should be done to promote the application of AAS. IC with SAPs was utilized to mitigate the shrinkage of AAS mortar in this work. An experimental research was carried out to evaluate the effect of SAP dosage on the setting time, autogenous shrinkage, and compressive strength of AAS mortars. Scanning electron microscopy (SEM) and mercury intrusion porosimetry (MIP) were carried out to study the change of micro pore structure of mortars. It was found that the AAS mortar’s autogenous shrinkage was significantly mitigated at a low SAP content (0.05–0.2%), and the setting time of freshly mixed mortar was delayed to a certain extent. The relationship between SAP dosage, compressive strength, and autogenous shrinkage was established through the experiment and it is found that 0.2% SAP content is the optimum choice. AAS mortars modified by SAP are very promisingly applied in practical engineering considering their physical, mechanical, and financial performance, which can provide references for the wider application of AAS.

## 2. Materials and Methods

### 2.1. Raw Materials

The materials used in this experiment includes sand, NaOH solution, granulated blast furnace slag (GBFS) and SAP. GBFS has a specific surface area of 1.535 m^2^/g. The partical size and chemical composition of GBFS were measured by Mastersizer 2000 (UK Malvern Instruments Co. LTD, Malvern, UK) and X-ray fluorescence (XRF) (Rigaku Corporation, Nanjing, China), respectively and the results are shown in Figure 1 and Table 1. The mean size of GGBS is 20 µm. Crosslinked sodium polyacrylate named SAP-30 with a price of $3.8/kg was used as the IC agent purchased from Shandong Yousuo Chemical Technology Co. LTD (Linyi Lvsen Chemical Industry Co. LTD, Shandong, China). The density of SAP is 0.5–0.8 g/cm^3^. Figure 2 presents the SEM image of SAP, suggesting that the particle sizes of SAP range from 300 μm to 500 μm. Existing studies [23,34] have indicated that SAPs with sizes of 300–1000 µm display better performance in mitigating the autogenous shrinkage of concrete. Its water absorption rate was tested as high as about 530%. Analytically pure flake sodium hydroxide was purchased from Tianjin Hengxing Chemical Reagent Manufacturing Co., Ltd. (Tianjin, China) to prepare the activator, 7 M NaOH solution. Sand used in this paper was collected from Huaihe river, China whose fineness module and apparent density were 2.6 and 2650 kg/m^3^, respectively.

### 2.2. Mix Proportions

To investigate the effect of SAP on the autogenous shrinkage of AAS mortar, seven mixtures were considered by adding 0.05% to 0.5% SAP, which can be further found in Table 2. For the control mixture, its ratio of activator to binder (A/B) is 0.5. Normally, IC uses water reservoirs to carry water as the IC agent to mitigate the autogenous shrinkage of OPC or AAS materials [15,30]. However, in AAS system activated by alkali solutions, the released water from reservoirs may dilute the concentrations of the activators in the surrounding pores [35], leading to negative effects on the hydration of AAS. Based on this point of view, in this paper, the activator was absorbed by SAP to implement IC for AAS mortars. An additional activator was determined and added to ensure that all mixtures displayed a similar flow [36], as shown in Table 2.

### 2.3. Test Methods

#### 2.3.1. Setting Time

According to ASTM C191-19 [37], the setting time of AAS paste was determined by a Vicat instrument (Zhejiang Geotechnical instrument Manufacturing Co. LTD, Zhejiang, China). Average results of each batch of mixtures was reported by testing triplicate specimens. In order to prevent the influence of water evaporation on the surface of the pulp, the surface of the pulp should be covered with a plastic film throughout the process [38].

#### 2.3.2. Autogenous Shrinkage Test

The autogenous shrinkage of AAS mortars, $\varepsilon_{autogenous}$, was measured according to ASTM C1698-19 [39]. The fresh mixture was slowly poured into the corrugated mold with a length of 420 mm during vibration in four equal layers. A fast operation was carried out due to the flash setting of AAS mortars. After being cleaned with water, these molds were placed on waved plates and cured under 23.0 ± 1.0 °C. The initial length of the sample was measured at the time of final setting $t_{fs}$ using dilatometer bench equipped with a digital gauge at one end [35] as shown in Figure 3. The resolution of the digital gauge is 1 μm. For each batch of mixtures, triplicate samples were tested, and the average results were reported. The autogenous strain of the specimen at time *t*, expressed as µm/m, is calculated according to formula (1).
(1)$$\varepsilon_{autogenous}=\;\frac{L\left(t\right)-L\left(t_{fs}\right)}{L\left(t_{fs}\right)}\;\times \;10^{6}\;\mu m/m$$
where:

*L*(t) = the length of the specimen at time _t_, mm$t_{fs}$ = the time of final setting, when the first length measurement is performed, min.

#### 2.3.3. SEM and MIP

Flex1000 SEM with an acceleration voltage of 15 kV was used to examine the microstructure of the AAS mortars. Both polished and fractured samples were prepared, and gold particles were sputtered for 120 s to gain electric conductivity. The fractured samples were prepared by fracturing the mortar samples. S0 and S20 mortars at 28 days were examined by SEM.

The pore structures of the AAS mortars with different SAP dosage were evaluated by an AutoPore IV 9500 mercury intrusion porosimetry (MIP). S0, S5, S30, and S50 mortars at 28 days were examined.

The samples used for MIP and SEM analysis were collected after the compressive strength test and sequentially soaked in anhydrous ethanol to stop further hydration.

#### 2.3.4. Compressive Strength Test

AAS mortars were cast into specimens with the size of 50 mm × 50 mm × 50 mm according to ASTM C109/C109M-2b [40] using plastic molds and were cured in standard condition at 23 °C and RH > 95%. The dye-300 digital pressure tester produced by Cangzhou Wanxiang Instrument Equipment Co., Ltd. (Cangzhou, China) was used in the experiment with the accuracy class of I. Each batch contained triplicate samples with 0.5 kN/s and average results were reported.

## 3. Results and Discussion

### 3.1. Setting Time

Figure 4 presents the setting time of the AAS pastes. It was seen that the addition of SAP significantly prolonged the initial setting time of AAS pastes. With the increase of SAP dosage, the initial setting time was delayed from 67 min of S0 to about 90 min of internally cured mixtures. Additionally, the final setting times of internally cured AAS pastes were also found to be increased by about 10 min. Such increased setting properties of internally cured AAS pastes by SAP suggests that SAP released the entrained NaOH at very early age which promoted the ratios of activator to binder. Considering that AAS normally set quickly, the prolonged setting time by adding SAP may provide more time available for AAS mixtures to be mixed, delivered, and pumped in practical application.

### 3.2. Autogenous Shrinkage

The autogenous shrinkage of AAS mortars is shown in Figure 5. It was shown that without SAP, AAS mortar suffered excessive autogenous shrinkage almost linearly during the testing period. The recorded autogenous shrinkage at 28 days is as high as 1750 µε, leading to great risks of cracking. Nevertheless, once adding SAP into AAS mortars, the autogenous shrinkage was greatly mitigated or even eliminated as a result of IC. By adding 0.05% or 0.1% SAP, AAS mortars developed 4–6 times less autogenous shrinkage than that of S0. When the SAP dosage increased to 0.2% or more, the autogenous shrinkage was completely eliminated and AAS mortars took place slightly desired expansion. The recorded 28 day autogenous shrinkages of S20, S30, S40, and S50 are 43.65, 149.2, 196.8, and 266.7 με, respectively. This confirms the high efficiency of IC by means of SAP on mitigating the autogenous shrinkage of AAS mortars. On the mechanism of IC, the carried NaOH solution by SAP was driven out into the surrounding pastes by the differences of internal relative humidity. The saturation degree as well as the internal relative humidity of capillary pores is retained in high levels, contributing to reduced capillary tension and autogenous shrinkage [32]. Therefore, through adopting IC, a further step can be achieved before applying such a green binder as AAS in practice without autogenous shrinkage.

In addition, the autogenous expansion of internally cured AAS mortars was twice noticed by previous researchers. Sakulich and Bentz [33] attributed such expansion of internally cured AAS mortars to the production of expansive Si-rich gel. Chen et al. [35] confirmed the speculation of Sakulich and Bentz and further pointed out that the geopolymer reaction between perforated cenospheres and the surrounding alkalis may also produce additional expansive products. This autogenous expansion is of significant importance to further inhibit other kinds of shrinkage of AAS mortars, i.e., drying shrinkage.

### 3.3. Microstructure Analysis

The microstructure of AAS mortars with and without SAP was examined by both SEM and MIP, as presented in Figure 6 and Figure 7. Without SAP, as shown in Figure 6a, several cracks were found in the microstructure as marked by red dotted lines as a result of excessive autogenous shrinkage. For AAS mortars with 0.2% SAP, S20, Figure 6b suggested that SAP was dispersed well in AAS mortars and some cracks can be observed in this Figure 6c to further investigate the microstructure of a typical pore created by SAP. Flake NaOH crystals was found in this void along with some alveolate substance in Figure 6d, which may be dehydrated SAP remnants. The residual NaOH within pores may be initially released from SAP and will promote the hydration degree of AAS in long age, contributing to better mechanical strength. Figure 6e showed the surrounding of an SAP pore. A great number of zeolite-like products was produced in this area. They may further fill the microstructure and contribute to a denser AAS mortar.

Figure 7 shows the MIP results (cumulative pore volumes vs. pore size) of S0, S5, S30, and S50 at 28 days. The porosities of AAS mortars was found to be directly proportion to the SAP dosage [26]. Beyond that, the pore volumes of AAS mortars on the whole scale were also enlarged. This decreased the density of AAS mortars, causing a reduction of compressive strength.

### 3.4. Compressive Strength

The mechanical strength or compressive strength in particular of internally cured concrete is highly concerned since the application of IC normally sacrifice the compressive strength of concrete [32]. Figure 8 presents the compressive strength of AAS mortars with different amounts of SAP. This indicated that the compressive strength of AAS mortars decreased accordingly with the addition of SAP. The higher the dosage of SAP, the less compressive strength. The reductions of the compressive strength of AAS mortars with different amounts of SAP were 50–70% at 3 days, 50–70% at 7 days, 50–70% at 28 days and 50–70% at 56 days. However, the S10 specimens had a distinct response when compared to other specimens as well as to same specimens at 3, 7 and 28 days. This is because the porosities caused by 0.05% SAP are fewer than that of 0.1% SAP but the additional activator loaded by 0.05% SAP cannot meet the demand of later hydration of AAS mortar. The additional activator loaded by 0.2–0.5% SAP can promote hydration of AAS mortars compared to 0.1% SAP but the hydration production cannot fill most of porosities caused by a great deal of SAP. Such drawback of adding SAP into mortars has been widely reported in other references [27]. The swollen SAP generally has sizes of several hundred µm. Therefore, during the IC process, large voids are generated when the loaded water or solution by SAP has been released, causing reduction of the compressive strength. There is another reason that contributes to the mechanical losses of the mortars: the effect of capillary porosity induced by the presence of the hydrophilic polymer. SAPs have a strong tendency to absorb water and therefore their presence in the matrix will involve greater dosages of water than S0 mixtures. The higher water content promotes the formation of capillary porosity which negatively affects the mechanical properties. Conversely, when hydrophobic fillers are used (e.g., rubber aggregates) the trend is opposite. In fact, due to their dimensional and physico-chemical properties, non-polar aggregates reduce the water content in the mixtures, resulting in a lower porosity degree [41].

To correlate the loss rate of compressive strength with of AAS mortars SAP dosages, linear fitting was applied as shown in Figure 9. At each age, the relationship between the loss rate of compressive strength and SAP dosages are such that:(2)$$y_{3d}=141.727x+7.174$$
(3)$$y_{7d}=118.550x+5.106$$
(4)$$y_{7d}=121.584x+5.106$$
(5)$$y_{56d}=133.140x-0.310$$
where *y* is the loss rate of compressive strength and *x* is SAP dosage.

The reductions of the compressive strength of AAS mortars with SAP may be predictable and can be reduced by prolonging the curing ages. Considering the results of both the compressive strength and autogenous shrinkage, S20 is regarded as an optimal mixture with slight expansion and 15% decrease of 56 days compressive strength.

### 3.5. Feasibility of Application in Practical Projects

All alkali-activated materials suffer to excessive shrinkage during their service life. However, autogenous shrinkage, one of the biggest barriers to applying AAS mortars in practical projects, can eliminated by adding 0.2% SAP according to the results in this paper. Thus, about 736 g SAP (0.2% of slag) is needed for one ton of AAS mortar costing about $2.8, which is a very small proportion of the total cost, but the autogenous shrinkage of AAS mortars can be completely eliminated and the 15% loss of compressive strength is acceptable. The reduction of the compressive strength of AAS mortars with SAP may be predictable according to the Equations (2)–(5) for different curing ages. Moreover, the addition of SAP significantly prolonged the initial setting time of AAS pastes. Considering the very short setting time of AAS, the prolonged setting time by adding SAP will be helpful for the wide practical application of AAS. The AAS mortar modified by SAP can be used in highways, airfields runways, especially in masonry blocks and prefabricated units.

## 4. Conclusions

This paper investigated the effect of IC by means of SAP on the autogenous shrinkage and compressive strength of AAS mortars. Conclusions are drawn based on the experimental results.

(1) SAP displayed a high efficiency in mitigating the autogenous shrinkage of AAS mortars. By adding 0.05% or 0.1% SAP, AAS mortars developed 4–6 times less autogenous shrinkage than that of S0. When the dosage of SAP was more than 0.2%, the autogenous shrinkage was eliminated and AAS mortars expanded slightly ranging from 43.65 με to 266.7 με.

(2) The setting time of AAS pastes were prolonged by SAP. The initial and final setting time of AAS pastes increased about 30 min and 10 min respectively. Such prolonged setting time can offer more time for the mixing, delivering, and pumping of AAS concrete.

(3) The addition of SAP towards to mitigating the autogenous shrinkage of AAS mortars sacrificed their compressive strength. The reductions of the compressive strength of internally cured AAS mortars increased with the increased of SAP dosage. S20 has been regarded as an optimal mixture with a slight expansion and 15% decrease of 56 days compressive strength.

(4) SAP is a good IC agent to eliminate the autogenous shrinkage of AAS mortars. SAP was found disperse well in the microstructure of AAS mortars and more zeolite-like products would be produced. However, the addition of SAP significantly increased the porosities of AAS mortars.

(5) AAS mortars modified by SAP is very promisingly applied in highways, airfield runways, and especially in masonry blocks and prefabricated units, considering the physical and mechanical performance discussed in the paper.

## Figures and Tables

**Figure 1 materials-13-04318-f001:**
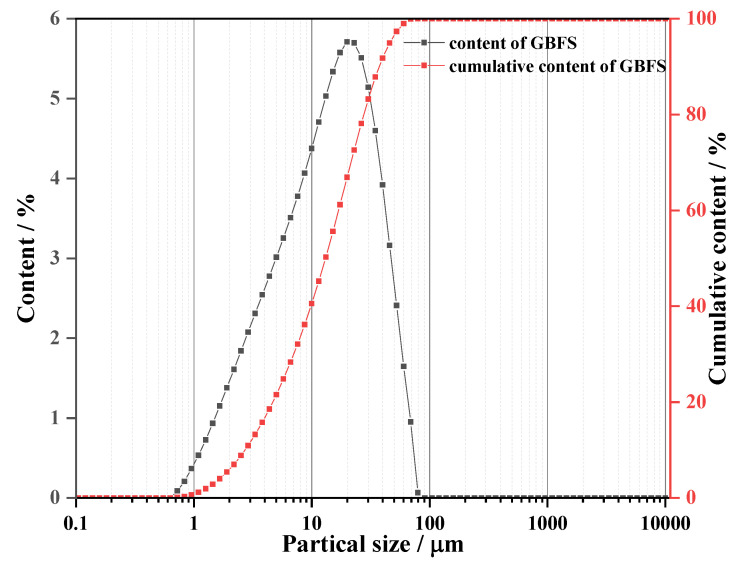
Particle sizes distribution of GBFS.

**Figure 2 materials-13-04318-f002:**
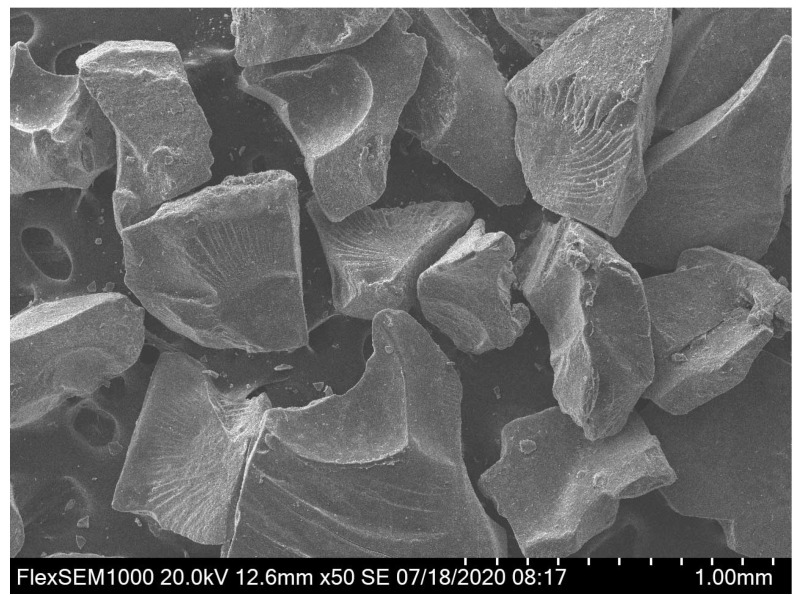
SEM image of SAP powders.

**Figure 3 materials-13-04318-f003:**
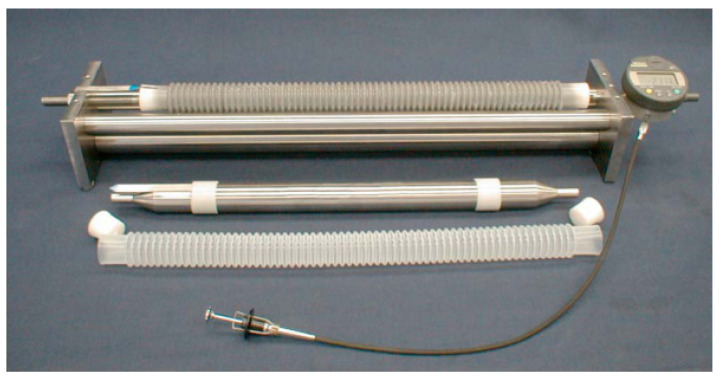
Schematic diagram of autogenous shrinking instrument [39].

**Figure 4 materials-13-04318-f004:**
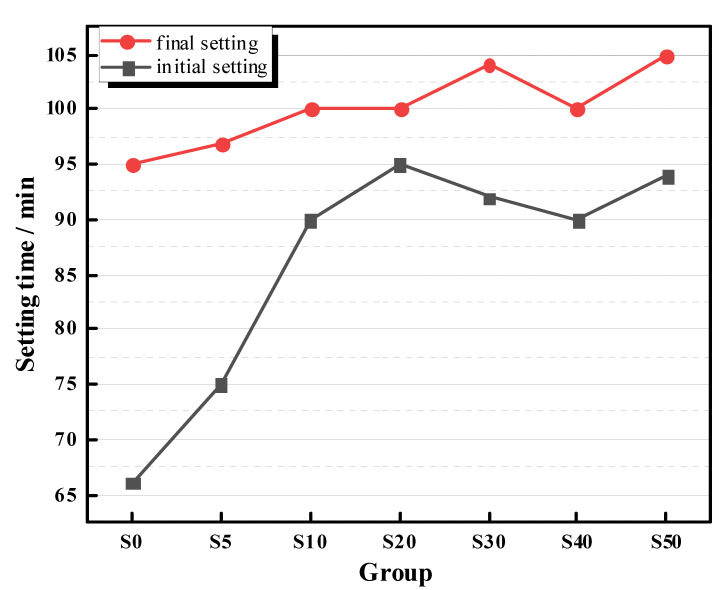
The setting time of AAS mortars with different dosages of SAPs.

**Figure 5 materials-13-04318-f005:**
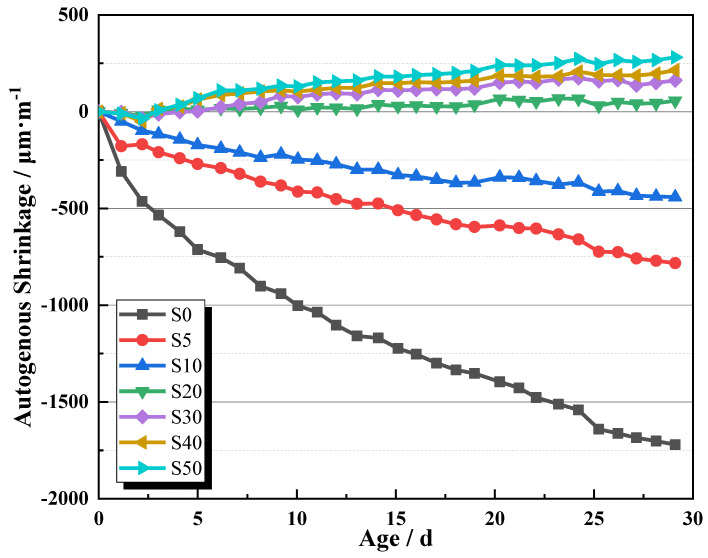
The autogenous shrinkage of AAS mortars as a function of curing time 3.3. Microstructure analysis.

**Figure 6 materials-13-04318-f006:**
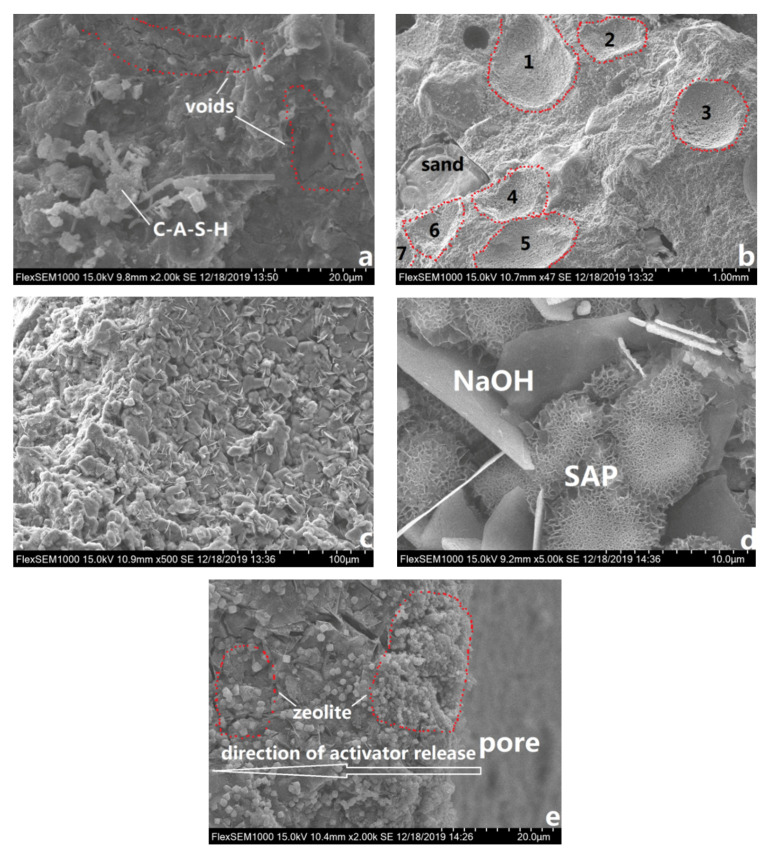
SEM images of AAS mortars: (**a**) S0 and (**b**–**e**) S20.

**Figure 7 materials-13-04318-f007:**
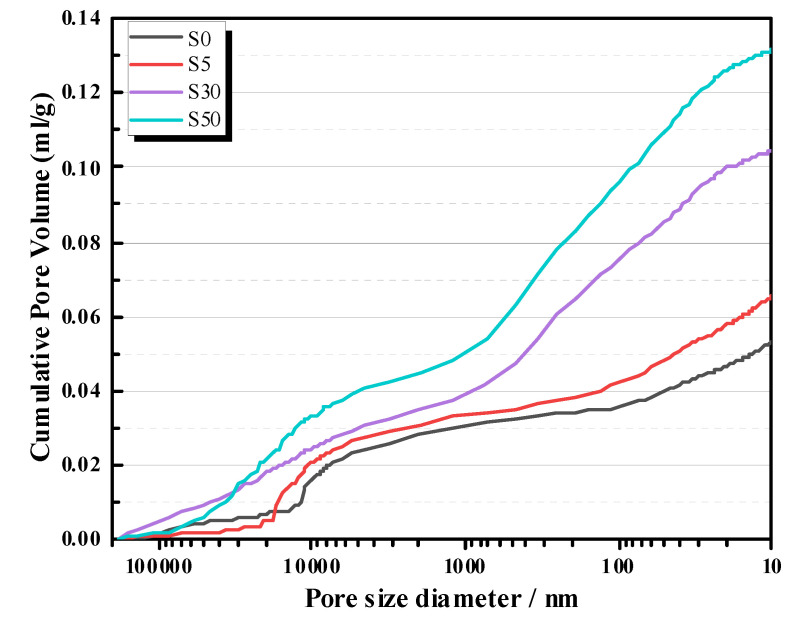
MIP of AAS mortars at 28 days.

**Figure 8 materials-13-04318-f008:**
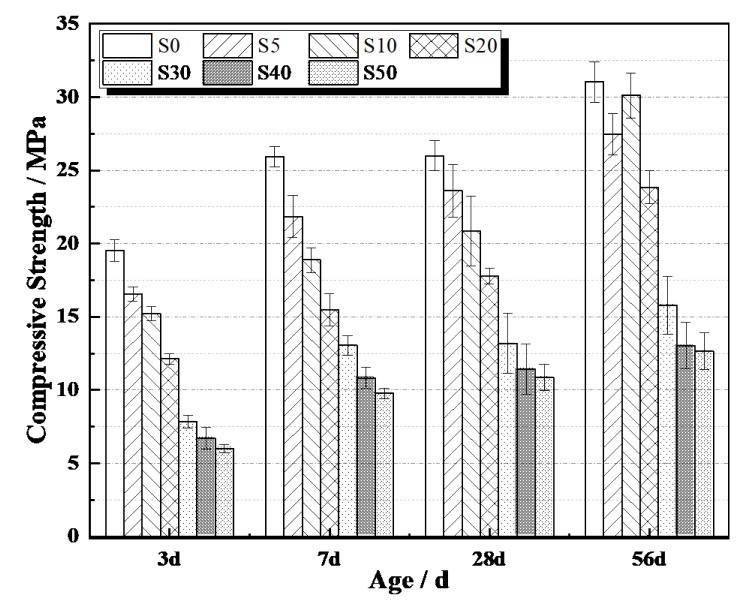
Compressive strength of AAS mortars.

**Figure 9 materials-13-04318-f009:**
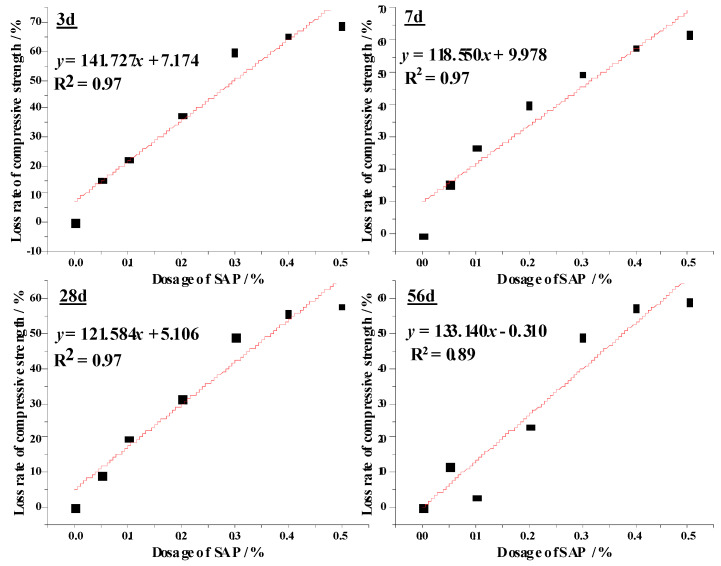
The linear fitting for the loss rate of compressive strength and SAP dosages.

**Table 1 materials-13-04318-t001:** The main chemical element composition of GBFS (wt./%).

CaO	SiO_2_	Al_2_O_3_	MgO	Fe_2_O_3_	TiO_2_	K_2_O
43.7	26.5	18.2	4.9	1.0	1.0	0.8

**Table 2 materials-13-04318-t002:** The mix proportion of AAS mortars (g/kg).

Sample	GBFS	Sand	SAP	Activator	Additional Activator
S0	367	450	0	183	0
S5	367	450	0.184	183	11.04
S10	367	450	0.368	183	22.08
S20	367	450	0.736	183	44.16
S30	367	450	1.104	183	66.24
S40	367	450	1.472	183	88.32
S50	367	450	1.840	183	110.40

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
