# Peer review of "Effect of Internal Curing by Super Absorbent Polymer on the Autogenous Shrinkage of Alkali-Activated Slag Mortars"

_materials, 2020, doi:10.3390/ma13194318_

Round 1

Reviewer 1 Report

Thank you for this contribution. This is an interesting and timely manuscript. This paper discusses how super absorbent polymer can be used to enhance autogenous shrinkage of AAS mortar. The conducted analysis is typically standard and falls within the expected work from such a publication and hence the work merits publication. As such, the authors are invited to properly address the following items:

  • The amount of works in this area continues to rapidly rise. The authors are advised to strengthen their literature review section with supplementary material. For example, the following is a result of a quick google scholar search and the authors may option to conduct their won search as well.
    • Dang et al. (2017) Polymers MDPI
    • Chehab et al. (2020) Polymer Science and Innovative Applications
    • Yang et a. (2019) Construction and Building materials
  • A key item to consider is the following. The authors have a good study with a good potential for large applications. However, the importance and novelty of this work is not properly highlighted. The authors are encouraged to better articulate their work. 
  • In Fig. 3, what is the reasoning behind achieving a reduced setting time at S30 and S40 as opposed to S20 and S50.
  • What is the expected rise in cost associated with the use of the proposed polymer?
  • In Fig. 5, the S10 specimens seem to have distinct response when compared to other specimens as well as to same specimens at 3, 7 and 28 days. Please explore the reasoning for this unique change. 

Author Response

Dear reviewer,

Thank you for your comments concerning our manuscript entitled “Effect of internal curing by super absorbent polymer on the autogenous shrinkage of alkali-activated slag mortars” (materials-930341). Those comments are all valuable and very helpful for revising and improving our paper, as well as the important guiding significance to our researches. We have studied the comments carefully and have made corrections which we hope meet with approval. 

Attached file is the Response to your Comments.

We tried our best to improve the manuscript and made some changes in the manuscript. These changes will not influence the content and framework of the paper.

We appreciate for your warm work earnestly, and hope that the correction will meet with approval.

Reviewer 2 Report

This paper presents an experimental study on the effect of internal curing by superabsorbent polymers  for mitigating autogenous shrinkage of alkali-activated slag mortars. Furthermore, the structural and physico-mechanical properties of the AAS-based materials were investigated as a function of the SAP dosage.

In the last five years a lot of research papers have been published on the same topic, justifying an average degree of novelty of the present study. The structure of the manuscript is fine and the experimental results are presented adequately. However, several technical changes to the text are necessary and some discussions of the experimental results need to be improved to enhance the significance of this document.

Below, the authors can find my comments and suggestions.

  • Line 13: Please, use capitilization for "Portland". 

  • Line 18: Please, revise the sentence. In my opinion, the percentage relationship between SAP and slag is not clear

  • Line 19: What do the authors mean with "working"? Maybe they refer to "workability"? In fact, the workability is related to the setting time. Please, revise this part.

  • Line 72: Please, specify the meaning of "SEM" and "MIP" before using abbreviations in the text

  • Lines 80-81: The particle size distribution and chemical composition of GBFS are shown in Figure 1 and Table 1, respectively. Are these data evaluated by experimental methods or are they provided by the product supplier? If experimentally determined, specify the experimental methodologies used.

  • Lines 84: Specify the microscope's features and the imaging  parameters (Magnification, HV, SE or BSE mode) related to the acquisition of the SEM images.

  • Table 1: Please, revised the table content. The sum of the GBFS components is not 100%

  • Line 119: What are the technical specifications and the model of the dilatometer used in shrinkage test?

  • Line 121: In Equation 1, the authors define the autogenous strain with "εautogenous". This nomenclature should also be mentioned earlier in the text.

  • Line 127: Please, provided information about the test machine' features and  test parameters (load speed, load cell...)

  • Line 136: What are the technical specifications and the model of the MIP used for porosimetry analysis?

  • Line 179: In my opinion, the discussion on  the mechanical strength reduction of AAS mortars is incomplete. The authors attribute this trend solely to the formation of large voids related to the effect of the SAPs size. There is another reason that contributes to the mechanical losses of the material: the effect of capillary porosity induced by the presence of the hydrophilic polymer. SAPs have a strong tendency to absorb water and therefore their presence in the matrix will involve greater dosages of water than S0 mixture. The higher water content promotes the formation of capillary porosity which negatively affects the mechanical properties. Conversely, when hydrophobic fillers are used (eg rubber aggregates) the trend is opposite. In fact, due to their dimensional and physico-chemical properties, non-polar aggregates reduce the water content in the mixtures, resulting in a lower porosity degree. It is suggested to report this discussion, inserting a reference to support this aspect (see: 10.3390/recycling5020011). The increase in porosity is demonstrated in the MIP results below. The authors should then highlight the difference in the water dosage between the investigated samples (consider the mix proportions reported in Table 2).

  • Lines 193-194-195-196: The authors report the fitting equations for each investigated age. However, it would be correct to specify which condition each equation refers to. For example, use a subscript for "y" variable indicating the curing age (y3d, y7d....)

  • Line 206: Microstructural analysis should be discussed before mechanical characterization to better correlate the reduction in compressive strength with the increase in porosity degree

  • Figure 7: SEM images have poor resolution and the notes inside are hardly visible, especially in 7a, 7c, and 7d images

  • Additional suggestion: To confer greater originality and scientific value to the paper, it might be interesting to dedicate a section on the possible technological applications of these mortars, considering the physical and mechanical performance discussed in the manuscript.

Author Response

Dear reviewer,

Thank you for your comments concerning our manuscript entitled “Effect of internal curing by super absorbent polymer on the autogenous shrinkage of alkali-activated slag mortars” (materials-930341). Those comments are all valuable and very helpful for revising and improving our paper, as well as the important guiding significance to our researches. We have studied the comments carefully and have made corrections which we hope meet with approval. 

Attached file is the point-to-point response to your Comments.

We tried our best to improve the manuscript and made some changes in the manuscript. These changes will not influence the content and framework of the paper.

We appreciate for your warm work earnestly, and hope that the correction will meet with approval.

Round 2

Reviewer 1 Report

Thanks for your efforts.